# Lightweight Multi-Scale Dilated U-Net for Crop Disease Leaf Image Segmentation

**Cong Xu, Changqing Yu and Shanwen Zhang ***

School of Electronic Information, Xijing University, Xi'an 710123, China
* Correspondence: zhangshanwen@xijing.edu.cn

**Abstract:** Crop disease leaf image segmentation (CDLIS) is the premise of disease detection, disease type recognition and disease degree evaluation. Various convolutional neural networks (CNN) and their modified models have been provided for CDLIS, but their training time is very long. Aiming at the low segmentation accuracy of various diseased leaf images caused by different sizes, colors, shapes, blurred speckle edges and complex backgrounds of traditional U-Net, a lightweight multi-scale extended U-Net (LWMSDU-Net) is constructed for CDLIS. It is composed of encoding and decoding sub-networks. Encoding the sub-network adopts multi-scale extended convolution, the decoding sub-network adopts a deconvolution model, and the residual connection between the encoding module and the corresponding decoding module is employed to fuse the shallow features and deep features of the input image. Compared with the classical U-Net and multi-scale U-Net, the number of layers of LWMSDU-Net is decreased by 1 with a small number of the trainable parameters and less computational complexity, and the skip connection of U-Net is replaced by the residual path (Respath) to connect the encoder and decoder before concatenating. Experimental results on a crop disease leaf image dataset demonstrate that the proposed method can effectively segment crop disease leaf images with an accuracy of 92.17%.

**Keywords:** crop disease leaf image segmentation (CDLIS); U-Net; dilated convolution; lightweight multi-scale dilated U-Net (LWMSDU-Net)

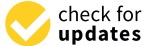



## 1. Introduction

Plant diseases severely affect the quality and yields of crops. Early detection of crop diseases reduces economic losses and has a positive impact on crop quality [1,2]. Crop disease leaf image segmentation (CDLIS) is a key prerequisite for the automatic detection, early warning, diagnosis and recognition of leaf diseases [3,4]. However, CDLIS is an important and challenging topic due to the various colors, shapes, textures, sizes and backgrounds of crop disease leaf images, as shown in Figure 1 [5,6].

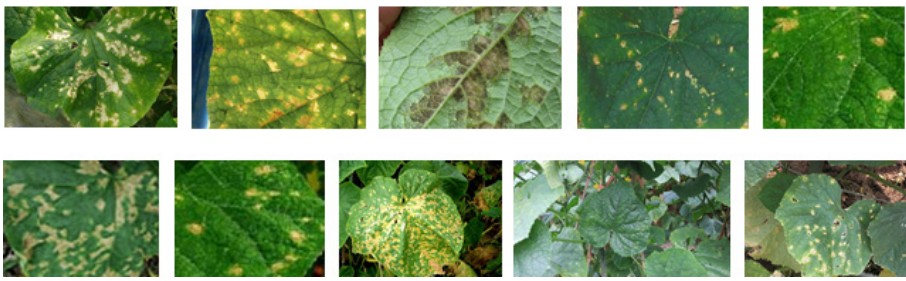

**Figure 1.** Disease leaf image examples.

Many image segmentation algorithms, such as fixed threshold, Otsu, K-means clustering, C-means clustering, fuzzy clustering, maximum entropy, 7 invariant moments and Local Binary Patterns (LBP), can be applied to CDLIS [7]. Wang et al. [8] proposed an

adaptive CDLIS method based on K-means clustering by three stages. Fernandez et al. [9] applied principal component analysis (PCA) to the spectrum to evaluate the spectral separability between healthy and infected leaves, used the spectral ratio between infected and healthy leaves to determine the optimal wavelength for disease detection, and applied the linear support vector machine (SVM) classifier to some spectral features.

The accuracy of the above traditional algorithms mainly depends on experience, and due to the complexity of diseased leaf images, they lack generalization ability. With the improvement of computing power, storage, Internet of Things, big data and artificial intelligence, deep learning methods, such as convolutional neural network (CNN), full convolutional neural network (FCN) and U-Net, have been widely applied to the detection, segmentation and classification of crop disease leaf images, and achieved a significant accuracy rate [10–13]. Ashwinkumar [14] proposed an optimal mobile network-based CNN (OMNCNN) for detecting and classifying plant leaf diseases. It involves bilateral filtering-based image preprocessing and Kapur's thresholding-based image segmentation to detect the affected portions of the leaf image. U-Net is a relatively simple and widely used image semantic segmentation model and has achieved remarkable performance in medical image segmentation. However, its segmentation performance for very multi-scale small targets may be poor. U-Net can be improved from many aspects, such as encoder number, convolution operation, up-sampling and down-sampling operation, residual operation, attention mechanism, multi-scale convolution, model optimization strategy and connection type between encoding and decoding layers [15,16]. Tarasiewicz et al. [17] proposed a lightweight U-Net (LWU-Net) and applied it to multi-mode magnetic resonance brain tumor image segmentation, obtaining accurate brain tumor contour. Xiong et al. [18] proposed a multi-scale feature fusion attention U-Net (AU-Net) to improve the defect detection accuracy caused by large background noise, unpredictable environments, and different defect shapes and sizes in defect images of industrial parts. This model combines attention U-Net with a multi-scale feature fusion module to detect the defects in low-noise images effectively. Yuan et al. [19] presented an improved AU-Net, which can integrate deep and rich semantic information and shallow detail information to perform adaptive and accurate segmentation of aneurysm images with large size differences in MRI angiography. Multi-scale U-Net (MSU-Net) can concatenate the fixed and moving images with multi-scale input or image pyramid and concatenate them with corresponding layers of the same size in U-Net [20]. Tian et al. [21] proposed a modified MSU-Net with dilated convolution structure, squeeze excitation block and spatial transformer layers. Experiment results indicated that it is competitive for normal and abnormal images. Wang et al. [22] proposed an improved U-Net namely HDA-ResUNet with residual connections, adding a plug-and-play, portable channel attention block and a hybrid dilated attention convolutional layer. It makes full use of the advantages of U-Net, attention mechanism and extended convolution, and performs accurate and effective medical image segmentation for different tasks. In U-Net, some related discriminant features may be lost in image segmentation.

Inspired by LWU-Net, AU-Net and MSU-Net, a multi-scale dilated U-Net (LWMSDU-Net) is constructed to improve the performance of CDLIS. It is lightweight, and the dilated convolutional coding operation is used to fuse features from different sizes of receptive fields. The main contributions of this paper are as follows:

- LWMSDU-Net is constructed by retaining local and multi-scale detail information;
- Dilated convolution is introduced into U-Net to enlarge the receptive field of the convolution layer, improve the feature learning ability of U-Net, and obtain more information about leaf spot image;
- A residual path (Respath) connection instead of the skip connection is employed to allow gradient information to flow better through the network and overcome gradient vanishing and degradation.

The rest of this paper is arranged as follows. Section 2 introduces the related works. LWMSDU-Net is described in detail in Section 3. A lot of experiments are conducted on a

crop disease leaf image dataset in Section 4. Finally, the paper is concluded and the future work is given in Section 5.

## 2. Related Works

### 2.1. Residual Block

The difference between the residual convolution block and the standard convolution block is that there is a skip connection [23]. Skip connection can effectively reduce the problems of gradient vanishing and network model degradation. Residual is the difference between the predicted value and the observed value. Suppose the first layer of the network is described as $Y = H(x)$, and a residual block of the residual network is noted as $H(x) = F(x) + x$, then $F(x) = H(x) - x$, and $y = x$ is the observed value and $H(x)$ is the predicted value, $H(x) - x$ or $F(x)$ is the residual, so it is also called the residual network.

### 2.2. Dilated Convolution

The basic principle of dilated convolution is to fill 0 in the middle of the convolution kernel to expand the receptive field as a principle, which is shown in Figure 2. By setting different expansion rates for each layer, multi-scale convolution domains can be obtained, thus obtaining multi-scale features. Its advantage is that the receptive field is enlarged without loss of features by pooling, so that each convolution output contains a wide range of features. Figure 2a corresponds to a 1-dilated convolution of 3 × 3, which is the same as an ordinary convolution operation without filling 0. Figure 2b corresponds to a 2-dilated convolution of 3 × 3. The actual convolution kernel size is still 3 × 3, but the void is 1, that is, for a 7 × 7 image patch, only 9 red points have convolution operation with a 3 × 3 kernel, and the rest points are skipped. Figure 2c is a 4-dilated convolution operation reaching the receptive field of 15 × 15. Compared with the traditional convolution operation, when the convolution of 3 layers and 3 × 3 is added together, the stride is 1, and the receptive field can only reach (kernel-1) × layer + 1 = 7, that is, the receptive field of dilated convolution increases exponentially. The corresponding convolutional images are shown in Figure 3.

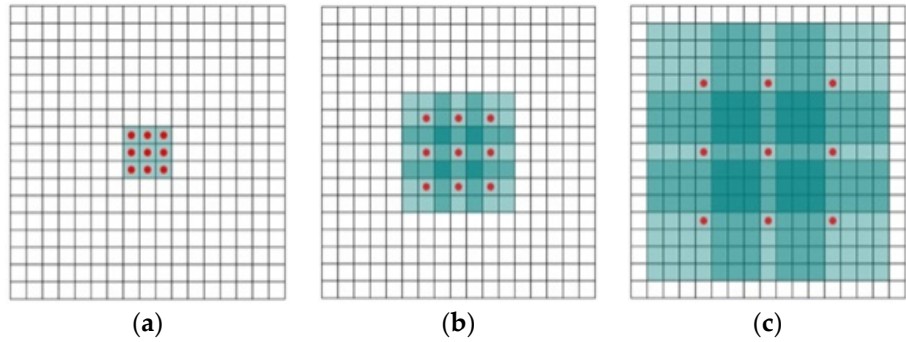

| (a) | (b) | (c) |

**Figure 2.** Dilated convolution kernel: (**a**) rate = 1; (**b**) rate = 2; (**c**) rate = 4.

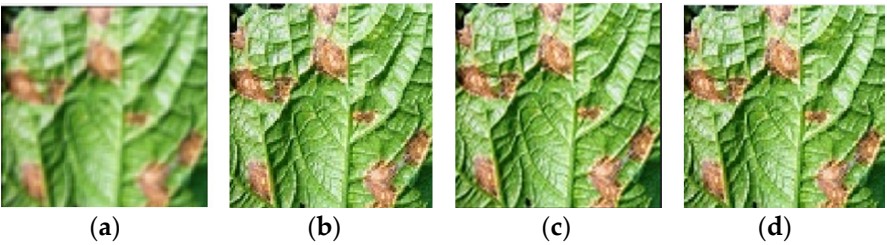

| (a) | (b) | (c) | (d) |

**Figure 3.** Dilated convolution images: (**a**) original image; (**b**) rate = 1; (**c**) rate = 2; (**d**) rate = 3.

### 2.3. U-Net

U-Net consists of a mutually symmetrical encoding subnetwork, decoding subnetwork and the skip connection. Its basic architecture is shown in Figure 4.

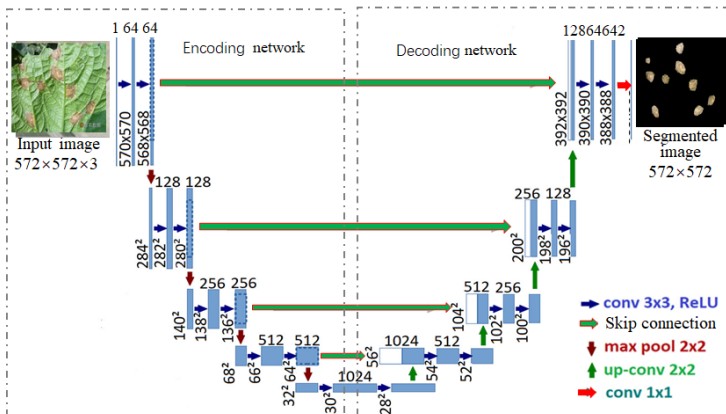

**Figure 4.** U-Net architecture.

Encoding subnetwork consists of four down-sampling operations and middle-layer operations, and each down-sampling operation includes Conv 3 × 3, BN, ReLU, MaxPool 2 × 2, DEA, where Conv 3 × 3 is 3 × 3 convolution for feature extraction, BN is a batch normalization layer to alleviate the problem of gradient disappearance, ReLU is the activation layer used to introduce nonlinear factors and accelerate network convergence, and MaxPool 2 × 2 is the maximum pooling layer of 2 × 2 to extract semantic information. Decoding subnetwork takes the output of the coding subnetwork as the input and carries out three upsampling operations, which are described as upconv 2 × 2 + Copy&crop + Conv 3 × 3 + BN + ReLU + DE module, where upconv 2 × 2 is a 2 × 2 upsampling convolution operation used to restore the size and size of the feature maps, and copy&crop, namely skip connection, refers to integrate the rough features of the encoding subnetwork with the refined features of the decoding to better retain the spatial information and detail information of the original image and then improve the image accuracy.

*2.4. Summarization*

The characteristics of the Residual block, dilated convolution and U-Net are summarized as follows.

Residual blocks can increase the depth of the network, help solve the problems of gradient disappearance and gradient explosion, and ensure good performance while training deeper networks.

When the network layer requires a large receptive field, but the computing resources are limited and cannot increase the number or size of convolution kernels, dilated convolution can be considered. Its advantages are that the receptive field can be increased without pooling information, so that each convolution output contains a large range of information. However, the dilated convolution may have a grid effect, that is, the convolution kernels are discontinuous; if only multiple 3 × 3 convolution kernels with dilation rate = 2 are stacked multiple times, not all input pixels are calculated. The key to designing a good dilated convolution layer is how to deal with the relationship between objects of different sizes at the same time.

U-net can provide context semantic information of segmentation target in the whole image, train end-to-end from a few images, and is superior to the previous sliding window convolution network. It uses features spliced together in the channel dimension to form thicker features, which can provide finer features for image segmentation. The addition of corresponding points used in FCN fusion does not form thicker features.

**3. Lightweight Multi-Scale Dilated U-Net (LWMSDU-Net)**

*3.1. LWMSDU-Net Architecture*

Although many improved U-Net models have been constructed and achieved remarkable results, they do not take into account the number of trainable parameters, the calculation of the model, and the characteristics of the disease leaf image shown in Figure 1,

and are not suitable for deployment on devices with limited computing power and storage space. To improve the accuracy and effectiveness of CDLIS, a lightweight multi-scale dilated U-Net (LWMSDU-Net) is constructed for CDLIS by making use of the advantages of lightweight, multi-scale, residual convolution, and dilated U-Net. Its architecture is shown in Figure 5a.

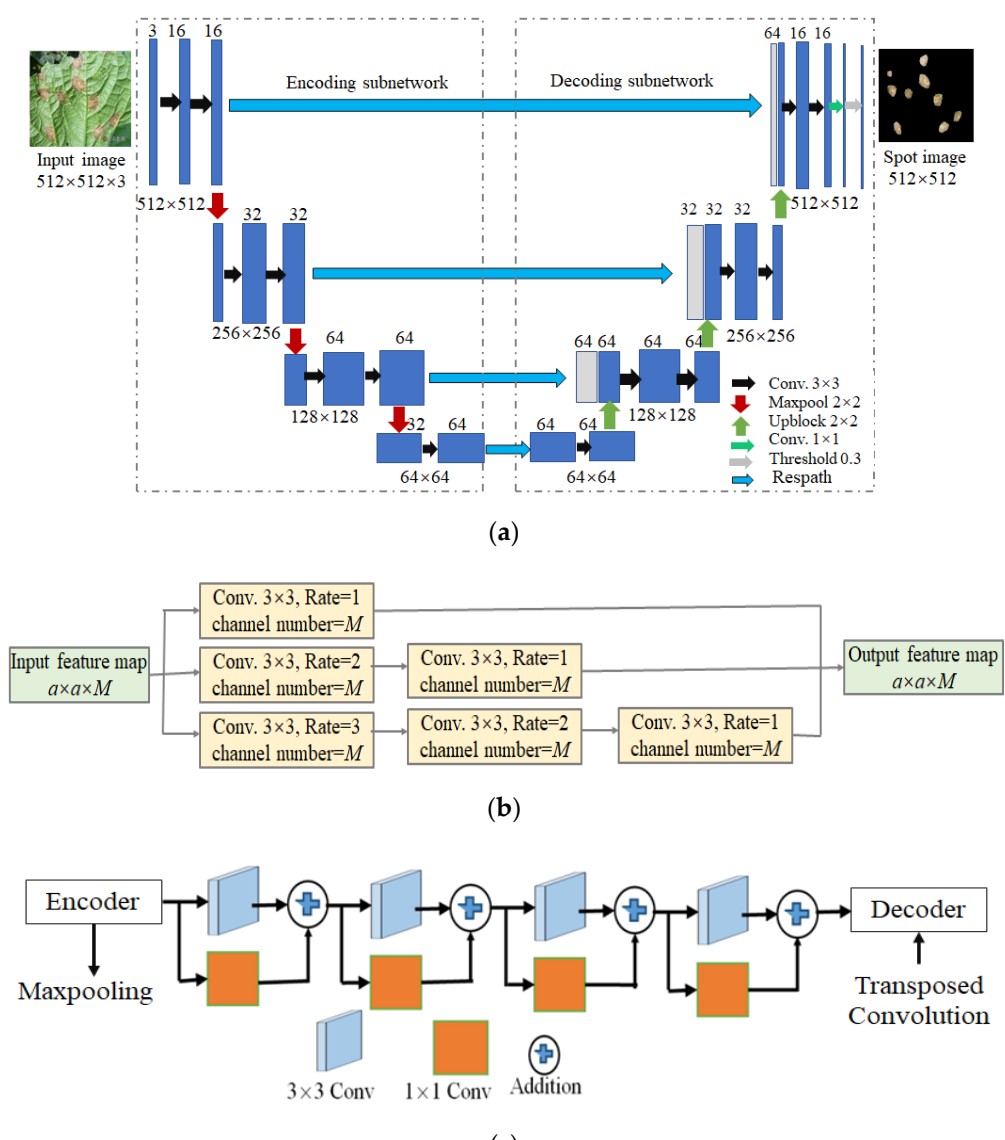

**Figure 5.** The architecture of LWMSDU-Net: (**a**) LWMSDU-Net structure; (**b**) dilated structure, where the size of the feature map is $a \times a$, and M is the channel number; (**c**) Respath structure.

The multi-scale dilated convolution is employed instead of the convolution of U-Net in the convolutional layer. According to the characteristics of the diseased leaf image, the expansion rates r is set as 1, 2 and 3, respectively, so that the irregular disease regions of different scales could be effectively segmented and the overall segmentation performance could be improved. The structure of multi-scale dilated convolution is shown in Figure 5b. Suppose $f_h$ represents the height of the original convolution kernel and $f_w$ is the width of the original convolution kernel, the height of the effective convolution kernel thus obtained is $f_h + (f_h - 1)(r - 1)$, and the width is $f_w + (f_w - 1)(r - 1)$.

In U-Net, the skip connection is used to connect the encoder and decoder. It is simple to implement, but there is often a big semantic gap in semantic between the encoder and decoder due to the complex disease leaf images. To improve segmentation results

and relieve this semantic gap, a residual path (Respath) instead of the skip connection is constructed to connect the encoder and decoder before concatenating, so that the encoder features perform some additional convolution operations before being spliced with the corresponding features in the decoder. Respath structure is shown in Figure 5c, consisting of four residual convolution blocks [22].

*3.2. Process of CDLIS*

The steps of LWMSDU-Net based CDLIS method include training stage and test stage. The original parameters of LWMSDU-Net are set by transfer learning, then the training dataset is used to optimize its parameters iteratively, and the test set is used to verify the model recognition effect. Model training is the most crucial step in the experiment because the trained appropriate model can improve the classification accuracy, and the experiment mode and hyper-parameter configuration of this paper are standardized to ensure the validity of the experiment. In the model training stage, to enhance the model image feature extraction ability and training speed, the PlantVillage dataset is used as the input of LWMSDU-NET, and the parameters of pre-training are retained. Then, the network model after pre-training is trained by the constructed augmented dataset of maize corn cucumber diseases. Pre-training can accelerate the model training speed, effectively enhance the fitting ability of the network, and improve the accuracy of CDLIS on the limited dataset.

The training stage includes the following steps:

Step 1: Convert the disease leaf images from R*G*B color space to L*a*b, and using the simple linear iterative cluster (SLIC) method to preprocess the transformed disease leaf images;

Step 2: Disease leaf images are converted to TensorFlow2 format, divided into different batches and then input into LWMSDU-Net for feature extraction (https://github.com/tzutalin/labeling/releases, accessed on 7 October 2022);

Step 3: Use transfer learning to reduce the number of training iterations and speed up training the network;

Step 4: Fuse the extracted features from LWMSDU-Net, and input the fused features into the classifier for training the classifier;

Step 5: If the error between the authentic labeled training images and the predictive labeled training images is more than the given threshold, go back to Step 2 and further train LWMSDU-Net. Otherwise, the training stage is stopped.

The test stage includes the following steps:

Step 1: Normalize the scale of the test images;

Step 2: Put the normalized images into the trained LWMSDU-Net and extract features;

Step 3: Fuse the extracted features and then put them into the SoftMax classifier;

Step 4: Output the recognition result of the input image.

## 4. Experiments and Analysis

In this section, a lot of experiments of CDLIS are conducted to validate the proposed method. Comparative experiments and results are then analyzed and discussed. All experiments are carried out: Windows 7 64-bit operating system, Intel Xeon E5-2643v3 @3.40 GHz CPU, 64 GB RAM, NVidia Quadro M4000 GPU, 8 GB of video memory, by CUDA Toolkit 9.0, CUDNN V7.0, Python 3.5.2, Tensorflow-GPU 1.8.0 with Keras open source deep learning framework. In LWMSDU-Net, the initial weight parameters are set randomly, the number of iterations is set as 500, the initial learning rate is specified as 0.001 and then gradually reduced to 0.1 times in training stages, the momentum is set as 0.99 to reduce the overfitting problem, the weight decay is set as 0.005, and the training images are divided into 10 batches and sent to the network model in turn. To improve the segmentation effect of the model, LWMSDU-Net is trained 1200 rounds with each round of iteration of 3000 times, and the widely used stochastic gradient descent (SGD) is used as a training mechanism. Since the last layer of the network is the Softmax classifier, Softmax-loss is

used as a loss function, which is more stable in computing. Other parameters are set as the default parameters of the U-Net framework. The trained model is evaluated by the verification images. In LWMSDU-Net, all RGB images of disease leaf are preprocessed through median filtering and then standardized by cropping to reduce calculation and training time. Each image is normalized and cropped to a size of $512 \times 512$ pixels.

### 4.1. Dataset

PlantVillage (https://tensorflow.google.cn/datasets/catalog/plant_village, accessed on 7 October 2022) is an open source dataset. It was collected at experimental research stations associated with Land Grant Universities in the USA (Penn State, Florida State, Cornell and others). It is an open source dataset for diagnosing and recognizing crop diseases. It consists of 54,303 healthy and unhealthy leaf images of 26 diseases of 14 crops taken in the natural environment of farmland. In this paper, it is utilized for pre-training to make up for the shortage of the training samples. The pre-trained model is then trained and tested using the real crop dataset.

In this paper, five types of maize and cucumber disease were taken with digital cameras, smart phones and other devices in the Yangling Agricultural Demonstration Field, Shanxi Province, including two corn leaf images of blight and brown spot, and three cucumber leaf images of target spot, brown spot and anthracnose, 20 leaf images for each disease. As the disease leaf images vary with crop growth environment, background, sunshine and photographic equipment, to reflect the real scene and improve the generalization ability of the model, all images were taken in the morning, noon, afternoon, sunny and cloudy days from April to June 2021. Five disease leaf image samples are shown in Figure 6.

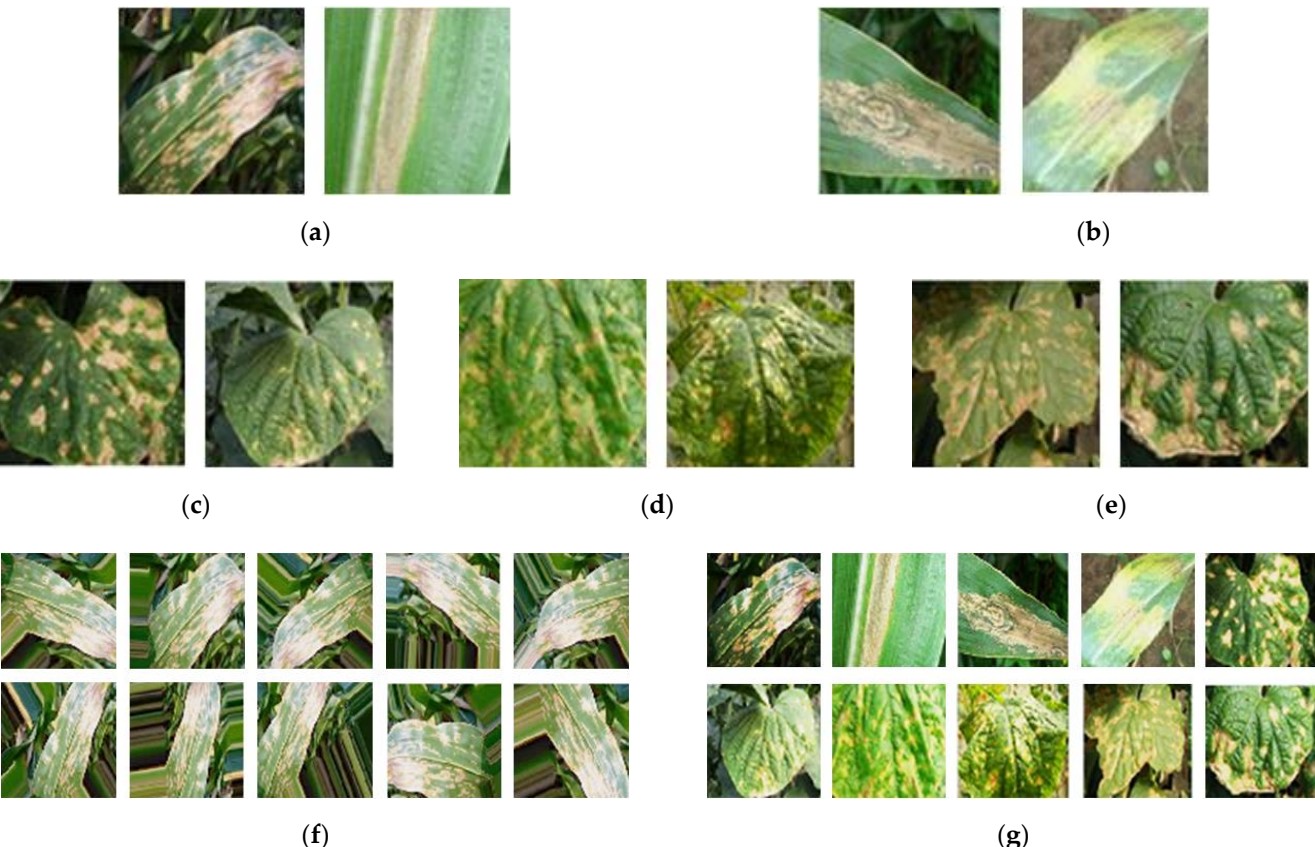

**Figure 6.** Five typical disease leaf images: (**a**) Original images for leaf blight of maize; (**b**) original images for brown blotch of maize; (**c**) original images for target spot of cucumber; (**d**) original images for brown spot of cucumber; (**e**) original images for anthrax disease of cucumber; (**f**) 10 augmented images of a maize disease leaf image; (**g**) equalized images of the above images in the above (**a**–**e**).

The number of the collected disease leaf images is limited, which easily leads to the overfitting. Augmenting algorithms, such as randomly enhanced lighting, randomly cropping, rotation, shifting, adding random noise and mirroring, are often used to augment the number of training samples. Augmenting operation can enlarge the diversity of the training samples and avoid overfitting. In the following experiments, each image is augmented to 10 images, as shown in Figure 6f. An augmented dataset containing 1100 images is constructed, including 100 original and 1000 augmented images. The details of the original dataset and its augmented dataset are shown in Table 1.

**Table 1.** The details of the original dataset and its augmented dataset.

| Disease Type | | Number of Original Images | Number of Augmented Images | Total |
|---|---|---|---|---|
| Corn | Leaf blight | 20 | 200 | 220 |
| | Brown spot | 20 | 200 | 220 |
| | Target spot | 20 | 200 | 220 |
| Cucumber | Brown spot | 20 | 200 | 220 |
| | Anthracnose | 20 | 200 | 220 |
| Total number of images | | 100 | 1000 | 1100 |

To reduce environmental noise and computational complexity, smooth the image, remove salt and pepper noise and retain image edge information, the median filtering algorithm is carried out on the crop disease leaf image, as follows:

$$y(n) = med[x(i-N), \ldots, x(i+N)] \tag{1}$$

where $x(i)$ is the value of the pixel point in the center of the sliding window, *med* is the value of the pixel's neighborhood, and $y(n)$ is the median filtering output value.

From Figure 6g, it is observed that median filtering can enhance the contrast of the disease leaf images and the filtered images can significantly characterize the disease leaf image features. The image recognition accuracy of CDLIS can be improved after median filtering.

The effective disease leaf image blocks are cropped from the collected images to reduce the influence of complex background on CDLIS, and the leaf images are uniformly processed into 512 × 512 resolution images. Secondly, Labelme is used to label the image set of crop disease leaves in the demonstration base. Each image contains two data labels: 1 represents the area of crop leaf disease spots, and 0 represents the background. Annotation data are stored in JSON format, and the command of labelme json to the dataset is used to convert data labels into binarized PNG graphs. The color annotated image can be obtained by multiplying the original and binarized images. The cropping and annotating process is shown in Figure 7.

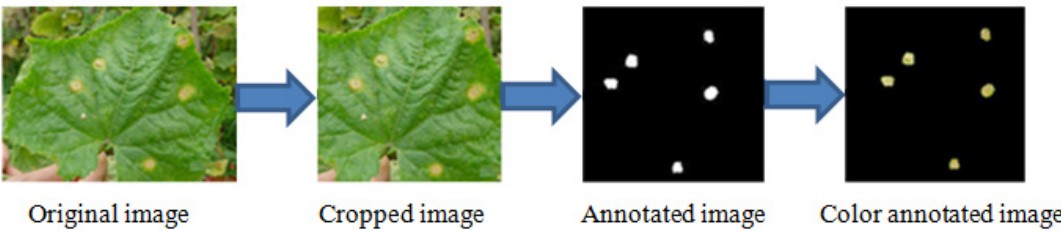

Original image　　　　Cropped image　　　　Annotated image　　　　Color annotated image

**Figure 7.** The cropping and annotating process.

In order to reduce the influence of geometric transformation and accelerate the speed of gradient descent to find the optimal solution, each image is normalized, which is implemented by mapping the pixel value of the image to (0,1) by linear function transformation,

$$y = (x\text{-MinValue})/(\text{MaxValue-MinValue}) \tag{2}$$

where $x$ and $y$ are the values before and after conversion, respectively, and MaxValue and MinValue are the maximum and minimum values of the sample, respectively.

There are some methods to form the statistical tests [24]. In the paper, a five-fold cross verification scheme is employed to validate LWMSDU-Net, that is, all 1100 leaf images per disease are randomly divided into five subsets with the same number of images, each is used as a test set for testing the model, and the remaining images are used as training samples for training the model. Each subset is taken as a test set once, and a total of five tests are conducted. The average segmentation result of five times experiments is the final result.

### 4.2. Results

Average precision, average recall and average $F_1$-score of five-fold cross verification experiments are often adopted to test network performance, and are calculated as follows:

$$Recall = \frac{B_{seg}}{B_{seg} + I_{unseg}} \tag{3}$$

$$Precision = \frac{B_{seg}}{B_{seg} + I_{wseg}} \tag{4}$$

$$F_1\text{-score} = 2 \times \frac{precision \times recall}{precision + recall} \tag{5}$$

where $B_{seg}$ is the pixel number correctly segmented into spot pixels, $I_{unseg}$ is the pixel number not segmented into spot pixels but being spot pixels in the image, and $I_{wseg}$ is the pixel number that segments the background pixels into spot pixels.

Pixel accuracy $Acc_{Pixel}$ is often used to evaluate the performance of the model. It is the total number of pixels whose real pixel category is predicted as a category, which is calculated as follows,

$$Acc_{Pixel} = \frac{1}{m} \sum_{i=1}^{m} f_i, f_i = \left\{ \begin{array}{l} 1, \left| y_i - y_i' \right| < T \\ 1, \left| y_i - y_i' \right| \geq T \end{array} \right. \tag{6}$$

where $y_i$ is the $i$th real pixel category, and $y_i'$ is the $i$th predicted category, $T$ is a threshold.

In fact, the final output of the image segmentation models is a grayscale image and the values of all pixels vary from 0 to 1, $T$ is often set 0.5.

LWMSDU-Net is trained on the augmented dataset. The training accuracy and loss are recorded after each iteration, as shown in Figure 8. It can be seen from Figure 8 that with the increasing number of training iterations, the accuracy of the model keeps rising while the loss value keeps decreasing. When the number of iterations reaches 2500, the accuracy is stable at 0.91, the fluctuation is stable within 1 percentage point, the loss value is stable at 0.043, and the fluctuation is within 0.01. The model has high accuracy and good robustness. It can be observed from the analysis that the LWMSDU-Net in this paper is effective and feasible for CDLIS.

The pre-trained model on the PlantVillage dataset is trained in the constructed dataset. In order to test the training performance of LWMSDU-Net, it is compared with U-Net, LWU-Net [17], AU-Net [18] and MSU-Net [21] on the augmented dataset. Each of the three improved models has its advantages, where LWU-Net is a lightweight U-Net, AU-Net takes advantage of attention, and MSU-Net is a multi-scale U-Net. Figure 9 shows their segmentation accuracies versus the number of iterations in the convergence process, where all models are pre-trained on PlantVillage dataset. From Figure 9, it is observed that all loss values of five network models drop rapidly before the 1000th iteration, and are nearly stable after the 1500th iteration. From Figure 9, it is also found that LWMSDU-Net outperforms other four models and achieves the best convergence performance after the 2700th iteration. The reason may be that dilated convolution and Respath are used to speed up its training and improve its segmenting performance. Comparing Figures 9 and 10, it can be found that the performance of LWMSDU-Net after pre-training is very good.

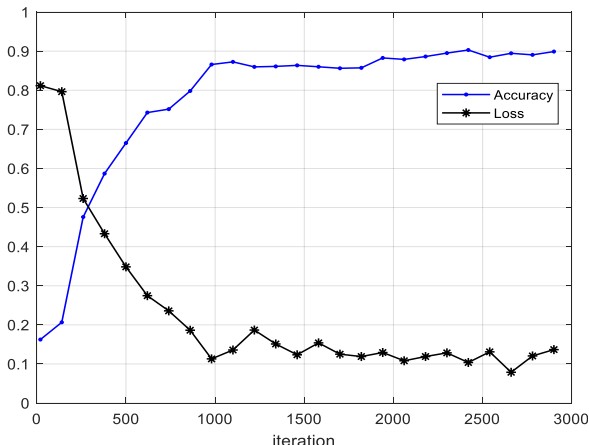

**Figure 8.** Accuracy and loss value versus iteration.

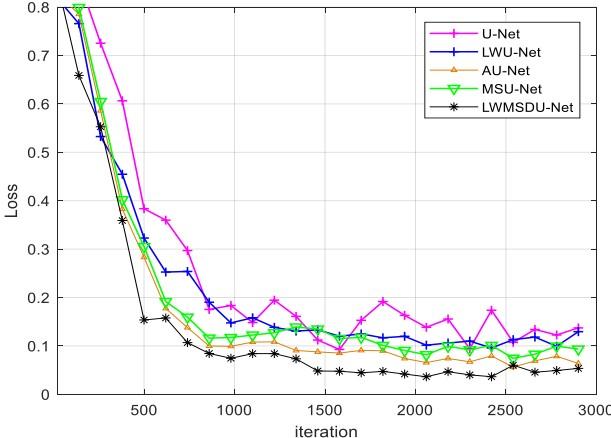

**Figure 9.** Segmentation accuracy versus the number of iterations of four networks.

To be fair, 5 trained models are chosen after the 3000th iteration. The typical segmented disease leaf images of five models are shown in Figure 10.

From Figures 9 and 10, it is observed that all four modified U-Net models are much better than the traditional U-Net. In five-fold cross verification experiments, the trained U-Net, LWU-Net, AU-Net, MSU-Net and LWMSDU-Net are used to segment the disease leaf images of the augmented dataset, and their segmentation results are shown in Table 2.

**Table 2.** Segmentation results of U-Net, LWU-Net, AU-Net, MSU-Net and LWMSDU-Net.

| Method | U-Net | LWU-Net | AU-Net | MSU-Net | LWMSDU-Net |
|---|---|---|---|---|---|
| Precision | 86.13 | 89.86 | 92.54 | 93.25 | 94.18 |
| Recall | 82.36 | 81.18 | 84.31 | 85.25 | 89.10 |
| $F_1$-score | 84.20 | 85.30 | 88.23 | 89.07 | 91.57 |
| Pixel accuracy | 85.66 | 90.24 | 91.50 | 91.45 | 93.71 |
| Training Time | 12.51 h | 6.42 h | 10.52 h | 11.14 h | 5.17 h |
| Testing time | 5.64 s | 5.18 s | 5.42 s | 4.85 s | 4.73 s |

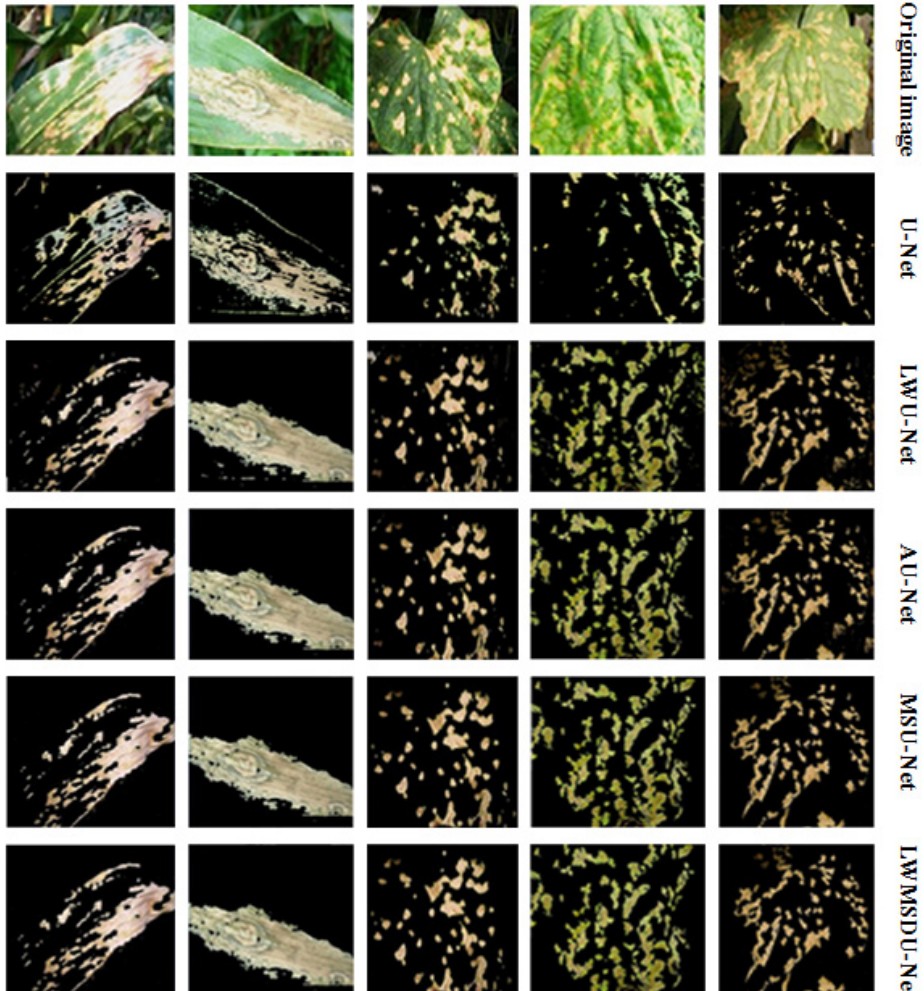

**Figure 10.** Typical segmented disease leaf images by 5 models.

### 4.3. Ablation Experiments and Results

The proposed model LWMSDU-Net is based on U-Net, and makes use of the characteristics of the Respath connection, dilated convolution and multi-scale Inception module. To verify the effectiveness of their combination, some ablation experiments are carried out. The experimental results are shown in Table 3 by combining different convolution structures and connection structures, where U-Net employs 3 × 3 convolution and skip connection, Res-U-Net is combined by U-Net and residual block for image segmentation [25], and Inception U-Net consists of a normalization layer, convolution layers, and Inception layers (concatenated 1 × 1, 3 × 3, and 5 × 5 convolution [26].

**Table 3.** Segmentation results by different combinations of convolution and connection.

| Combination Mode | Precision | Training Time |
|---|---|---|
| U-Net: 3 × 3 conv.+ Skip connection | 86.13 | 12.51 h |
| U-Net: 3 × 3 conv. + Respath connection | 87.22 | 11.36 h |
| Res-U-Net: residual block + Skip connection | 90.14 | 11.75 h |
| Inception U-Net: Inception + Skip connection | 92.16 | 10.46 h |
| U-Net: Inception module + Respath connection | 91.57 | 9.73 h |
| U-Net: dilated Inception module + skip connection | 92.46 | 7.13 h |
| LWMSDU-Net: dilated Inception + Respath connection | 94.18 | 5.17 h |

From Table 3, it is found that the proposed LWMSDU-Net exhibits quite significant results as compared to the original U-Net, Inception U-Net and different combination

architecture, and the results validate the effectiveness of dilated Inception module, Respath connection and their combination.

## 5. Analysis and Discussion

From Figures 9 and 10 and Tables 2 and 3, it is observed that LWMSDU-Net and other modified U-Net networks can obtain more detailed spot images even if the spots are small and not clearly contrasted with the healthy leaf areas and background, and specially, LWMSDU-Net is superior to the other models in accuracy and computing complexity. LWU-Net and LWMSDU-Net have shorter training times because they are lightweight and have fewer trainable parameters, while LWMSDU-Net has the shortest training time because it utilizes dilated convolution and Respath connection. U-Net splices the features together in the channel dimension to form richer segmentation features. U-Net can completely segment the lesion area including the small lesion area, but it cannot effectively divide the adhesion lesion, resulting in more missing lesion pixels. CDLIS by U-Net has some false positive areas, which could not distinguish the lesion area from the background. CDLIS by its modified models is better than that of U-Net. MSUN-Net is slightly better than LWU-Net and AU-Net due to the multi-scale convolution. AU-Net is slightly superior to LWU-Net because of the attention mechanism. LWMSDU-Net can accurately segment the disease leaf images including the lesion area and the edge details of the lesion, due to it utilizing Respath instead of the skip connection of U-Net, and dilated convolution instead of convolution. It is indicated that Respath and dilated convolution can improve the performance of CDLIS.

Compared with other networks, the experimental results demonstrate that LWMSDU-Net achieves a significant segmentation effect. However, it is validated only on a single enhanced dataset. The super-parameters of the training network need to be adjusted according to the dataset being processed, so it cannot completely guarantee that the model weight parameters can be transmitted to other data sets.

In terms of the memory occupied by the model, VGG16 occupies the largest memory, 552.0 MB, the memory occupied by AlexNet is 227.6 MB, because the number of parameters of the fully connected layers is the largest in the entire model. In deep CNN, in order to increase receptive fields and reduce the amount of computation, it is always necessary to conduct downsampling (pooling or s2/conv). In this way, although the receptive fields can be increased, the spatial resolution is reduced. In order not to lose resolution and still expand the receptive field, dilated convolution can be used. By adding zeros to expand the receptive field, the original $3 \times 3$ convolution kernel can have a $5 \times 5$ (dilated rate = 2) or a larger receptive field under the same parameter amount and calculation amount, so that no down sampling is required. Dilated convolution introduces only a parameter called dilated rate to the convolution layer, which defines the distance between values when the convolution kernel processes data. In other words, compared with the original standard convolution, the extended convolution has an additional division rate parameter. The division rate of a normal revolution is 1. It can be observed that the number of parameters of the dilated convolution is greatly reduced. Based on this, dilated convolution is added to U-Net, which effectively reduces the number of model parameters. The number of parameters of U-Net is 7.76 M, while the number of parameters of this model after training is 5.8 MB.

## 6. Conclusions

Aiming at the problem of crop disease leaf image segmentation (CDLIS), the traditional U-Net model is improved by making use of dilated convolution and Respath. Multi-scale dilated convolution instead of traditional convolution is used to increase the receptive field and improve the feature learning ability of U-Net. Respath instead of skip connection between Encoder and Decode is utilized to concatenate the lesion information of disease leaf image. PlantVillage is employed for pre-training to make up for the shortage of the training samples, overcome the overfitting problem and improve the network performance.

The proposed CDLIS method based on LWMDU-Net can be applied to the actual agricultural environment, help farmers quickly and accurately detect crop diseases, and provide effective technical means for scientific disease control. For future work, it is necessary to further verify and optimize the model and construct a more lightweight version for deployment to personal computers and smartphones.

**Author Contributions:** Conceptualization, C.X. and S.Z.; methodology, C.X.; software, C.X. and C.Y.; formal analysis, C.X. and C.Y.; writing—original draft preparation, C.X.; writing—review and editing, S.Z. All authors have read and agreed to the published version of the manuscript.

**Funding:** This work is supported by the National Natural Science Foundation of China (Nos. 62172338 and 62072378).

**Data Availability Statement:** Not applicable.

**Conflicts of Interest:** The authors declare no conflict of interest.

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
