# Peer review of "Lightweight Multi-Scale Dilated U-Net for Crop Disease Leaf Image Segmentation"

_electronics, doi:10.3390/electronics11233947_

Round 1

Reviewer 1 Report

The paper proposed a modified UNet structure for Image segmentation that focus on a specific application which is to localize crop diseases with images.

However the improvements are all based on already proposed work such as dilated convolution, residual connection and Unet. The work put these together without much scientifically convincing justification.

The experiment part needs comprehensive ablation study to demonstrate how adding dilated convolution and/or residual connection improves the performance.

Only one dataset with limited data is used to demonstrate the performance. The dataset description is confusing, it begins with 250 images of 5 diseases and then 100 images for each disease. One line 281 it says 1100 leaf images per disease. So how many images is there actually in the dataset?

Given the fact that the novelty of the paper, the dataset and code are also not openly accessible. I tend to think the value are not that appealing for the readers.

Reviewer 2 Report

  1. Enhance the introduction to show the motivation of this work.
  2. The manuscript organization should be improved.
  3. There should be some discussions on the limitations of the presented methods in a separate section. 
  4. References should be updated; there is only one reference in 2022.
  5. Enhance the English of the work. There are too many problems with paper typesetting.
  6. It is necessary to discuss the complexity of the proposed solution.
  7.   Performance evaluation metrics are not enough. Add some other metrics.

  8.  The proposed method should be compared with more recent techniques.

  9. Some statistical tests, such as ANOVA and T-Test, should be performed to ensure the quality of the proposed algorithm. you can see how statistical tests form this reference :

    Ibrahim, Abdelhameed, Seyedali Mirjalili, Mohammed El-Said, Sherif SM Ghoneim, Mosleh M. Al-Harthi, Tarek F. Ibrahim, and El-Sayed M. El-Kenawy. "Wind speed ensemble forecasting based on deep learning using adaptive dynamic optimization algorithm." IEEE Access 9 (2021): 125787-125804.
  10. Some additional experiments are required: Scalability and Runtime.

Reviewer 3 Report

1.       Abstract needs to refine by mentioning the limitations in the existing work and supposed to mention their work over them.

  1. In section 2 materials and methods, the authors must add the related works and their limitations in terms of either description way or summary table.
  2. Specify the details about the total Dataset and Training and test dataset in terms of size, attributes, features, etc.
  3. Abbreviate short forms when first used.Ex. SLIC
  4. Section 3.2, mentioned normalized the scale of the test images. How and when the images are normalized? Is it before training or after training?
  5. Authors mentioned that the features were extracted from the trained LWMSDU‐Net. List the features considered.
  6. Please clarify your method advantages more clearly.
  7. Suggested to consider the following works.

·         https://doi.org/10.3390/w14142234

·         https://doi.org/10.1007/978-981-19-2719-5_3

Round 2

Reviewer 1 Report

1) I appreciate the author for addressing most of my concerns properly, including making the dataset openly accessible. 2) However the experiment is lacking an ablation study of the impact of introducing dilated convolution and residual connections. I tend to consider this as a flaw in the experiment design. Thus remain my decision as major correction, since 1 table in the experiment part to demonstrate performance doesn't seem like a comprehensive analysis of the proposed model.

Reviewer 2 Report

Thanks for your review; well done this time

Show an overview of the importance of the main contribution of the proposed algorithm in point .

Some syntax errors or improper expressions exist in the manuscript.

Round 3

Reviewer 1 Report

I think the new version and new tables that demonstrate the ablation study is much better.